# The Brother–Sister Sibling Dyad as a Pathway to Gender-Based Violence Prevention: Engaging Male Siblings in Family-Strengthening Programs in Humanitarian Settings

Andrea Koris [1], Monica Giuffrida [2], Kristine Anderson [3], Hana Shalouf [4], Ibrahim Saley [5], Ahmad Marei [4], Ilana Seff [6], Julianne Deitch [2,*] and Lindsay Stark [6,*]

1 Independent Researcher, Cape Town 8001, South Africa
2 Women's Refugee Commission, New York, NY 10019, USA
3 Mercy Corps Headquarters, Portland, OR 97204, USA
4 Mercy Corps Jordan, Building No. 8, Tabasheer 3 Street, 7th Circle, Amman 11931, Jordan
5 REM Africa-Niger, Rue LZ0 Lazaret, Niamey P.O. Box 623, Niger
6 Brown School of Social Work, Washington University in St. Louis, St. Louis, MO 63130, USA
* Correspondence: julianned@wrcommission.org (J.D.); lindsaystark@email.wustl.edu (L.S.)

**Abstract:** Household violence poses a significant threat to the physical and mental health of adolescent girls. In conflict-affected communities, increased stresses to safety, security, health, and livelihoods may heighten this risk. While it is widely evidenced that the caregiver-child relationship can increase or protect against girls' risk of violence, less is known about the role of male siblings. Sibling Support for Adolescent Girls in Emergencies (SSAGE) used whole-family support programming to synchronously engage adolescent girls, their male siblings, and their caregivers in conflict-affected communities in Jordan and Niger, using gender-transformative approaches to explore the impacts of gender norms, power, and violence and encourage support and emotional connection. We conducted qualitative research activities, including focus group discussions, participatory group activities, and in-depth, paired, and key informant interviews with 469 SSAGE participants and program facilitators to explore SSAGE's impact on the male-female sibling dyad in both settings. The multi-stakeholder team used a collaborative thematic analysis approach to identify emergent themes. Findings suggest that the inclusion of male siblings in family strengthening programs may have a positive impact on factors related to girls' protection, with research participants discussing decreased perpetration of physical and verbal violence by male siblings, increased equity in household labor between siblings, and improved trust and mutual support among siblings. These changes were facilitated by improved communication and interrogation of positive gender identities. In humanitarian settings, interventions that support more gender-transformative, egalitarian, and emotionally effective relationships between male-female siblings can work towards improving girls' protective assets. More research on the impact of this relationship on girls' experience of immediate and long-term experience of violence is needed. In settings where gender power dynamics among male-female siblings are less salient, other relationship dyads should be explored.

**Keywords:** adolescent girls; family functioning; gender-transformative; mental health; well-being; refugees; conflict; household violence; gender-based violence; evaluation

## 1. Introduction

Recent global estimates suggest that nearly one in three women and adolescent girls have experienced violence at the hands of a male intimate partner, relative, friend, acquaintance, or stranger [1]. This statistic, however, falls woefully short of representing the true prevalence and scope of gender-based violence (GBV). The global burden of GBV is unequally shared between high and low- and middle-income countries, with the

latter experiencing disproportionately higher rates of GBV prevalence [2]. Moreover, in humanitarian contexts within those settings, the threat of GBV is further elevated [3–6].

Adolescent girls living in humanitarian settings are most at risk of GBV due to intersecting vulnerabilities related to their age, gender expression, and additional factors associated with complex emergencies. For girls living in these contexts, exposure to household-level violence, limits on mobility within community spaces, inequitable distribution of household labor, de-prioritization of educational attainment, and internalization of social norms which devalue women and girls can pose significant threats to their immediate and long term physical and mental health and wellbeing [7–14]. The negative impact of gender inequitable attitudes, norms, and practices on adolescent girls' mental wellbeing is particularly well documented [12,15,16]; as is the increased risk girls face of developing complex trauma and mental health sequelae following exposure to abuse, including depression, anxiety, self-harm, suicidal ideation and psychosomatic symptoms [14,17,18]. These dynamics are further evidenced in multi-country longitudinal studies such as Gender and Adolescence: Global Evidence (GAGE), which describes the association between gender inequitable community norms, family attitudes, reduced self-esteem, and wellbeing among adolescent girls in six middle- to low-income countries [12,19,20].

Evidence demonstrates that relational dynamics in adolescent girls' households can either protect them from or increase their risk of experiencing GBV and associated mental health sequelae in the immediate, mid, and long term. Households characterized by rigid gender and age hierarchies, where children are corporally disciplined or witness intimate partner violence (IPV) perpetrated against their female caregiver, can increase girls' risk of GBV, either through perpetration in the immediate term by male family members, or in the long term by future intimate partners [21–23]. Conversely, households where caregivers foster secure attachment with their children, encourage positive coping strategies, or promote egalitarianism and nonviolent communication between family members can function as a protective factor for adolescent girls against immediate or future violence victimization [8,24–26].

In the last decade, family strengthening programs—which work to address household-level risk factors, attenuate the psychological effects of trauma, and target intergenerational cycles of violence—have emerged as a promising approach for reducing adolescent girls' risk of GBV and supporting their mental wellbeing [27,28]. The majority of family strengthening programs, however, focus largely on the caregiver-child relationship and not on the relationship between siblings [29]. Although evidence suggests that sibling violence is highly prevalent, typically gendered in nature [30–35], and is associated with future violence victimization by peers [36], the male-female sibling dyad as a site of perpetration of or protection from violence against adolescent girls remains largely unexplored.

Most of the existing research on sibling relationships and violence is focused on high-income, non-humanitarian contexts in the Global North. Despite this, some of the major trends in the sibling violence evidence base are relevant for understanding this form of household violence in low-income, humanitarian contexts in the Global South. For example, studies conducted in Australia, the United States, and Portugal in the last ten years found that the majority of acts of sibling violence are committed by older boys against siblings who occupy inferior positions to them within sex/age hierarchies in the family; and that the occurrence of caregiver-directed violence against a child was strongly associated with onward perpetration of sibling violence [30–34]. In addition to learning violence within the household, various other studies demonstrate how violent masculinities are socialized amongst adolescent boys vis-a-vis peer violence perpetration and victimization both within school and community settings [37–39]. These trends illustrate that in certain contexts, sibling violence—particularly that between brother and sister—can function similarly to IPV in terms of its gendered nature, its relation to power, its connection to past and future experiences of violence for both perpetrator and survivor, and its role in the continued reproduction of dominant masculinities [40–42].

A theoretical model developed by Hoffman and Edwards in 2004 helps explain the phenomenon of gender-based sibling violence, including the use of physical, emotional, and psychological abuse, by integrating perspectives from feminist, conflict, and social learning theories. The model suggests possible drivers and pathways for gender-based sibling violence: (1) patriarchal norms in society and in families structure sibling relationships along rigid gender/age hierarchies, and brothers may perpetrate violence against their sisters for reasons which are normalized within their context or to assert masculinity; (2) male and female siblings may use verbal or physical violence to mitigate conflicts that stem from competing interests or resources (i.e., household labor and other familial responsibilities); and (3) siblings may learn and model violent gendered interactions from their caregivers, peers and the broader society they live in. Using this theoretical grounding, Hoffman and Edwards describe how sibling violence arises from the interactions between caregiver, caregiver-child, and sibling relationship characteristics, individual attitudes and characteristics, and siblings' verbal interactions [40].

*Sibling Support for Adolescents in Emergencies (SSAGE)* is a theory-driven program designed to disrupt violence pathways within households, improve mental health and resilience for adolescent girls living in conflict-affected settings, and strengthen relationships between brothers and sisters through enhanced emotional support, interpersonal communication, conflict management, and awareness of gender inequity. Findings from a qualitative process evaluation of a 2020 pilot of the program in northeast Nigeria suggested the SSAGE approach positively impacted the functioning of participating families, as well as individual attitudes and behaviors related to violence against women and girls [43,44]. However, additional research was needed to understand the distinct impact of the SSAGE approach on the male-female sibling dyad, and to explore the opportunities that strengthening male-female sibling relationships might offer for increasing girls' protection from interpersonal household violence. In 2022, Mercy Corps scaled up SSAGE to the Syrian refugee camps of Za'atari and Azraq in Jordan, and the Abala refugee camp for Malian refugees and its surrounding host communities in Abala, Niger.

Drawing on the theoretical grounding from Hoffman and Edwards, this paper uses qualitative data from mixed methods evaluations in both SSAGE program settings to explore the impacts of SSAGE on male-female sibling relationships in Jordan and Niger. In doing so, we focus on the following areas of inquiry: (1) what were the effects on sibling dyads following the SSAGE program in Za'atari, Azraq, and Abala, and what facilitated those changes, and (2) what were the unintended impacts of the program on sibling relationships?

## 2. Materials and Methods

### 2.1. Study Site: Syrian Refugee Community in Za'atari and Azraq, Jordan

There are approximately 670,000 Syrian refugees living in Jordan, of whom 18.7% live in the Za'atari and Azraq refugee camps [45]. Most households are composed of both nuclear and non-nuclear family members, many of whom are adolescents under the age of 18 [46,47]. Households hail from distinct regions of Syria, each with its own cultural, ethnic, and religious background (48% originate from Dara'a, 19% from Homs, 10% from Aleppo, and 9% from rural Damascus). While family structures are diverse and shaped by various factors from before and after displacement (i.e., class, ethnicity, religion, or rural-urban region of origin) [48,49], many of the norms governing familial relations stem from the mosaic of national Syrian religious and secular legal codes that uphold the centrality of male power and have governed Syrian social life since the 1950s [50,51]. The male-female sibling dyad often reflects this patriarchal structure and can serve as a site where masculinities and femininities are constructed and reproduced vis-a-vis each other [41].

In addition to restrictive gender norms at the household and community level, structural factors such as exposure to war and displacement, discriminatory laws, resource scarcity, unemployment rates as high as 23%, insecure housing, and poor access and quality of education also shape Syrian refugees' daily lives in Jordan. These factors contribute to

high rates of interpersonal violence [47]. For adolescent boys, lack of economic opportunity and pressures to contribute to household finances pushes them into informal work where they are at risk of physical violence or sexual abuse at the hands of older men from their community or the Jordanian host community [52]. For women and girls, sexual harassment in public spaces, violence at home, and IPV are particularly pervasive, but drastically underreported [53,54]. Child, early or forced marriage (CEFM) and removal from school are also major risks for Syrian girls ages 15–17 and are more common now than in pre-war Syria [47]; drivers of this include increased poverty levels due to displacement, low value placed on continuing education due to the reality of limited employment opportunities, protection against harassment and rape, and preservation of family honor [37,55]. Diminished political will from the global community to generate viable third-country resettlement options forces Syrian refugee families to adapt to new or exacerbated risks of violence that come with living in the refugee camp setting.

### 2.2. Study Site: Malian Refugee Community and Host Communities in Abala, Niger

Abala, a rural commune in Western Niger, is a highly fragile setting that is subject to extreme insecurity and displacement, militarization and the presence of armed groups, adverse climate events, food insecurity and malnutrition, and poor service infrastructure. Protracted regional conflicts between Islamist militant groups such as the État au Islamique au Gran Sahara, Boko Haram, and other affiliated splinter groups; secular armed groups such as the Touraeg political movement, Le Mouvement pour le salut de l'Azawad; and the Nigerien army fuel the internal displacement of Nigeriens as well as the forced migration of refugees from nearby countries including Burkina Faso and Mali. As of January 2019, the commune's refugee camp is home to approximately 15,684 Malian refugees, of which more than half are female and almost two-thirds are under the age of 18 [56].

The risk of violent attacks by armed groups, the heightened presence of the Nigerien military, the extended state of emergency in the Tillabéry region where Abala is located, and conflicts over resources between refugee camps and nearby host communities coincide to create an environment rife with high levels of community and interpersonal violence. For women and girls, the risk of experiencing sexual violence at the hands of conflict-related actors, family members, or people within their close social networks is high. Recent data show the prevalence of GBV that occurred in the Tillabéry region within the last 12 months for women and girls was 14.3%, slightly above the national prevalence of 13% [57,58]. Older national data show that almost one-third (28.4%) of women reported ever experiencing GBV. In Niger, most reported incidents of violence occur at home, with 91.9% of reported incidents of sexual violence and 61.4% of reported incidents of physical violence taking place in the domestic space [58]. Women and girls in Niger are affected by a matrix of religious and customary laws and community-held norms in both the Malian refugee and Nigerien host communities that uphold male power, underpin patriarchal family structures, and normalize the use of violence against women and girls. CEFM is a particular problem, as economic vulnerability, extreme resource scarcity, and limited economic or educational opportunities combined with community-held norms result in 76% of girls being married or in first unions before the age of 18 [58].

### 2.3. SSAGE Intervention

The SSAGE program was developed, implemented, and evaluated via a consortium of partners including Mercy Corps and local research organizations in Jordan and Niger, the Women's Refugee Commission (WRC), and Washington University in St. Louis (WashU). The project employed a gender transformative pedagogy and a whole-family approach to foster interactive spaces where individual family members alongside their peers could interrogate normative beliefs and interpersonal dynamics that fuel violence against women and girls. This approach was operationalized among families by enrolling an adolescent girl, one of their male siblings, and their male and female caregivers into the program; facilitating synchronized age and gender-specific sessions with each cohort over

a 3-month period; and encouraging intra-familial discussion on weekly topics throughout the duration of the program. Four distinct but corresponding curricula were developed for each cohort and addressed topics related to gender, power, violence, interpersonal communication, positive emotional coping skills, and healthy relationships (please see [43] for more information on the SSAGE intervention, including curricula topics by each of the four cohorts).

### 2.4. SSAGE Program Implementation

Prior to the implementation of the program, Mercy Corps teams in Jordan and Niger led a human-centered design community consultation with groups of adults and adolescents recruited from Mercy Corps' other protection programs. During these activities, the SSAGE curricula that were piloted in Nigeria were contextualized to Jordan and Niger to ensure the topics and facilitation methods were acceptable and relevant to the participating communities. Due to limited levels of education and literacy among intervention participants in Abala, the SSAGE curriculum evolved into a didactic image-based modality.

Families were eligible to participate in the program if, in their household, they had a male and female caregiver, an adolescent girl aged 10–14, and an older male sibling between the ages of 15 and 19 (in Jordan) or a male sibling between the ages of 13 and 24 (in Niger). These criteria were determined to ensure that program participants included members of adolescent girls' families who may have decision-making and disciplinary authority over them. Mercy Corps teams in both sites identified eligible households and household members by mapping the study site communities using elements of the I'm Here approach [59], conducting home visits, and inviting families to participate in the program. A total of 264 participants were enrolled across two cycles of the program in Jordan, and 601 participants were enrolled in Niger.

Mercy Corps staff in both settings identified dynamic female and male community members to serve as program mentors and facilitate the SSAGE sessions in their community. The program mentors in each site completed two weeks total of pre-cycle training, and during program implementation, Mercy Corps staff conducted field visits to oversee their facilitation and offer constructive guidance. Mentors led separate sessions for adolescent girls, their male siblings, and male and female caregivers in a synchronized and concurrent manner from June to September and October 2021 to January 2022 in Jordan, and from July to October 2021 and January to April 2022 in Niger.

### 2.5. Evaluative Study Design

Qualitative data were collected in both sites, as part of mixed methods evaluations looking at the impact of the SSAGE program on key program outcomes including adolescent girls' mental health and wellbeing [60], family functioning, and attitudes towards GBV. Qualitative research activities included in-depth interviews (IDIs) and focus group discussions (FGDs) with male and female adult caregivers, and paired interviews (PIs) and participatory research activities (PARs) with adolescent girls and their male siblings. Caregiver activities (IDIs and FGDs) were facilitated using semi-structured guides with open-ended prompts to elicit reflection and dialogue on the effect of the SSAGE program on adolescent girls' mental health and wellbeing, sibling and caregiver relationships, overall family functioning, and gender norms and roles, as well as perceptions of barriers to program participation, areas for improvement, and thoughts on curricula content. Research activities with adolescents were facilitated using PIs and PARs, such as story completion or arts-based methods (like dance, storytelling, and drawing/family mapping) that were structured in part around vignettes depicting gender and age-relevant scenarios, in order to encourage reflection and dialogue among the age group on the topics noted above. In addition, 12 SSAGE program staff and mentors from Jordan and Niger were purposely recruited to participate in key informant interviews (KIIs) using semi-structured guides with open-ended prompts, which focused on their perceptions of the acceptability and impact of the curricula and pedagogical methods of the SSAGE program. All data collection

tools were translated and validated using cognitive interviewing and pilot testing, to ensure linguistic and cultural relevance to the communities in Azraq, Za'atari, and Abala (tools are available upon request).

Qualitative data were collected approximately one to two months after the completion of each program cycle. In both Jordan and Niger, qualitative participants were recruited by the Mercy Corps teams using criterion sampling based on factors such as gender, age, and program attendance rate. In total, 213 participants were selected in Jordan and 248 in Niger (see Table 1 for more information).

**Table 1.** Qualitative evaluation samples in Jordan and Niger.

|  | Jordan | | Niger | | Total |
|---|---|---|---|---|---|
|  | **Cycle 1** | **Cycle 2** | **Cycle 1** | **Cycle 2** | |
| Male and female caregivers | 63 | 30 | 60 | 37 | **190** |
| Adolescent girls | 39 | 26 | 53 | 23 | **141** |
| Male siblings | 35 | 20 | 50 | 21 | **126** |
| Program staff and mentors | 0 | 8 | 0 | 4 | **12** |
| **Total** | **137** | **84** | **163** | **89** | **469** |

*2.6. Data Collection*

WRC recruited two mixed-gender research teams experienced with social protection research with displaced communities for the evaluations in Za'atari, Azraq, and Abala. Prior to each data collection cycle, WRC and the directors of both local research groups co-facilitated qualitative methods training with the research assistants to familiarize them with the topic guides and the methods, as well as WRC's and Mercy Corps' research ethics, community accountability, and participant safeguarding mechanisms.

In both sites, research activities were led by gender-concordant facilitators in the respective local languages (Arabic in Jordan and Hausa and Tamashek in Niger) and completed two to three weeks following the end of the SSAGE program. With the completion of the qualitative activities, the research assistants listened to the audio recordings, consulted observational notes when relevant, and transcribed the recordings. For the Jordan data, an external translator unaffiliated with the evaluation translated the Arabic transcripts into English for analysis, and another Arabic-speaking staff member from WashU not involved in the data collection spot-checked transcripts against audio files to ensure accuracy of translation. For the Niger data, audio recordings were transcribed directly into French for analysis.

*2.7. Analysis*

Upon completion of the first cycle of qualitative data collection in both Jordan and Niger, analysis teams from the WRC, WashU, and the respective Mercy Corps country teams participated in virtual analysis workshops to collaboratively develop a codebook for their respective setting. During the workshops, the teams mapped out emergent connections between the deductive categories and inductive themes and began to identify a list of codes that represented major patterns in the data. Lead coders for each setting then developed preliminary codebooks, which the analysis teams piloted on subsections of data using the constant comparison method [61], and continued to iteratively adapt until finalized codebooks were reached. The lead coders in each setting then worked to establish coding consensus with one another, after which they coded the remainder of the qualitative data. The lead coders then reviewed the data for each code and drafted thematic memos that explored the nuances in themes and incorporated relevant quotations to ground the analysis in participants' own words. The codebook adaptation, co-coding, and memo-writing process were then repeated in Jordan and Niger with the completion of the second round of data collection.

Our qualitative data revealed similarities between sites in the self-reported prevalence of gendered sibling violence; the types of violence experienced or perpetrated; and the

types of changes to sibling relationships following their participation in SSAGE. As such, we opted to present findings from both sites together, despite the sociocultural, community, and structural factors underpinning household and sibling violence amongst the Syrian refugee community in Jordan and the Malian refugee and Nigerien host communities in Niger.

*2.8. Ethics*

Study procedures in Jordan were approved by the Health Media Lab institutional review board and the Jordan University of Science and Technology institutional review board. Study procedures in Niger were also approved by the National Ethics Committee For Health Research of the Niger Ministry of Public Health. Participants in both sites provided written, informed consent and assent. Participants were provided with an information sheet with details related to support services for voluntary referral to specialized psychosocial support from Mercy Corps and partner organizations in case of possible emotional distress or protection risks associated with research study activities.

## 3. Results

The qualitative findings suggest that the SSAGE programs in Jordan and Niger successfully targeted several pathways of sibling violence within participating families and had a positive impact on factors associated with girls' wellbeing and protection from violence, including (1) decreased reported perpetration of corporal and verbal violence by male siblings, (2) improved perceptions of equity in the division of household labor between male and female siblings, and (3) increased emotional involvement and social support from male siblings to their sisters. These changes were described to have been facilitated by male siblings' interrogation of positive masculinities and improved communication between male and female siblings. Qualitative findings also suggest that, following the SSAGE program, male siblings reconceptualized their role in their sisters' safety in diverse ways, some of which prioritized restricting girls' mobility as a means of protection. Overarching themes from both sites that emerged in the course of analysis are organized as follows: (Section 3.1) changes in the sibling dyad associated with girls' increased protection from violence, (Section 3.2) pathways of change in the sibling dyad, and (Section 3.3) unanticipated impacts of male siblings' heightened awareness to protection risks faced by their sisters.

*3.1. Changes to the Sibling Dyad Associated with Girls' Increased Protection from GBV*

3.1.1. Decreased Physical, Psychological, and Verbal Abuse by Male Siblings

Throughout the various qualitative activities conducted with adolescent girls in Jordan and Niger, it was clear that the majority of girls regularly experienced a range of physical, verbal, and sometimes psychological abuse at the hands of their brothers. Qualitative story vignettes often led girls to reflect and share stories of their day-to-day lives, many of which laid bare the commonplace occurrence of brothers using verbal and physical aggression when attempting to control their sisters' behavior, force them into complying with their requests, and/or externalize their own negative emotions.

Following SSAGE participation, however, many adolescent girls remarked on the profound change in their brothers' behaviors towards them, and as a result, on the improved quality of their sibling relationship. A girl living in a host community in Abala, Niger shared the following: "*Really this program has affected my relationship with my brother in a good way because before between us it's war, we fight all the time but now we have all understood. We have become real brother and sister.*" (Adolescent girl from a paired interview, Abala). Another girl in a host community in Abala shared a similar sentiment: "*He stopped doing a lot of things. He beat me up, humiliated me, but he doesn't do that anymore because of this SSAGE program he's had. He does things right.*" (Adolescent girl from a paired interview, Abala).

In both settings, adolescent boys and male youth also reflected on their past instances of physical and verbal abuse towards their sisters and talked at length about how their participation in the SSAGE program heralded a positive change in their lives and relationships. When asked during a participatory activity to share stories of hypothetical characters who underwent programs like SSAGE, several male siblings living in Azraq, Jordan described the evolution of a character named Walid—a young man who no longer bullies and abuses his sister to get what he wants but instead works to listen, understand her plights, and build a relationship with her. They shared the following:

*"Walid was a guy who bullied his family, friends, and relatives. He listened to no one. When the program started, he wanted to participate. He registered himself and his relationship with his family and friends started to improve. They were surprised by the changes he was going through. He became understanding and his actions changed . . . his relationship with his sister was based on violence. He ordered her and forced her to do things. Now he helps her and doesn't treat her like a servant. Once he asked her for a glass of water and hit her when she didn't bring it. After the program though, they started to understand each other. The relationship changed."* (Male sibling from a participatory research activity, Azraq).

In the qualitative activities, some male siblings went beyond the hypothetical and spoke about changes that they made in the treatment of their sisters following SSAGE participation. An adolescent boy in Azraq shared how instead of shouting at his sister and ordering her to serve him as soon as he arrives home, he now focuses on greeting his sister, talking with her, and showing her respect: *"I used to come from school very hungry and enter the house without showing any respect or even saying hello. I used to start shouting the moment I entered the house and give orders to bring me food quickly. After the course, I enter the house respectfully and I listen to my sister and talk to her respectfully."* (Male sibling from a paired interview, Azraq).

Male siblings in Niger also reflected on the detrimental impact of their negative communication with their sisters and the ways in which their engagement has since changed. A male sibling living in a host community in Abala, for example, reflected on how the SSAGE program helped him find new ways of relating to his siblings. Instead of inspiring fear as a mode of coercion and control, he began taking up some of the interpersonal communication skills covered in the program. With this change, he in turn has discovered a more positive role for himself as an older brother, one who engages with his siblings and provides them with advice: *"My participation in the program has helped me a lot and I have learned a lot of things. For example, before the program I didn't talk to anyone at home, my sisters and brothers were afraid of me, I hit them, we didn't talk about anything, but after my participation now I stay at home to talk with my brothers and sisters, to advise them . . . "* (Male sibling from a paired interview, Abala).

The changes in male siblings' use of physical and verbal abuse against their sisters were corroborated by male and female caregivers, although overall female caregivers were often more forthcoming in their observations of this change. A mother from Azraq shared how SSAGE gave her son the skills to manage challenging emotions instead of externalizing them upon his younger sister: *"My son used to get angry easily. He used to talk to his sister in a nervous way. He used to take it out on her. Now this thing has changed. He leaves us when he is angry and tells us to stay away from him. We leave him alone till he is calm and then he apologizes if he did something wrong."* (Female caregiver from an in-depth interview, Azraq). A mother in Abala came to a similar understanding, sharing how her children stopped taking out their negative emotions on each other following SSAGE participation: *"We had a change between our children (girls and boys). Before they manifested insane behaviors between them. Now, they understand each other very well because they followed the training of"* SSAGE." (Female caregiver from a focus group discussion, Abala).

3.1.2. Increased Equity in Household Labor between Male Siblings and Their Sisters

Following SSAGE participation, some adolescent boys and male youth reported taking on a larger share of their family's chores to increase the equity of household labor between themselves and their sisters. This in turn seemed to have a positive impact on their sisters' wellbeing. A male sibling living in a camp in Abala, for example, described how the intervention helped him think more critically about the assumed gendered nature of household labor in his community, the ways that negatively impacted young women like his sister, and how he as a young man and older brother could choose to act differently:

*"There are many beneficial things that we have learned together. For example, we think that there are many jobs that are only for girls, but SSAGE has made us understand that these jobs are not only for men. Another example is that your sister stacks, prepares the meal . . . you can see that she has jobs, but "SSAGE" has taught us that you can reduce all these jobs that your sister does."* (Male sibling from a paired interview, Abala).

Similarly, an adolescent boy living in Za'atari, Jordan discussed how his decision to contribute more equally to the household labor allowed his sister more time to go to school: *"I [now] help her with the chores, I wash the dishes and arrange the room. She used to take the day off when she had to do the chores. I help her now, and she goes to school."* (Male sibling from an in-depth interview, Za'atari). Another boy in Za'atari shared how he now takes on his sister's chores when she has other events and obligations: *"A week ago, she wanted to visit her friend, it was her friend's birthday, and she had to clean the house too, so I told her that she should go and that I will clean the house."* (Male sibling from an in-depth interview, Za'atari).

Overall, adolescent boys and male youth from all program sites reflected that prior to SSAGE they used to "order [their sisters] and force her to do things" and now, they "help her with her chores, and they have stopped ordering them around." (Male sibling from a paired interview, Azraq). It is important to note that while SSAGE seemed to instigate shifts in how brothers thought about equitably sharing household labor with their sisters, data from other participant cohorts suggest that gendered labor equity within participating households was attenuated to varying degrees. Some boys gladly took over some of their sisters' household work, freeing up time for them to attend school or engage in social interactions. Other boys did not actively take on any of their sisters' chores but rather stopped expecting her to fulfill tasks in service of their needs: *"... when he sees that his sister or I are sick he will serve himself. His father tells him to assume that his sister is not there and that she doesn't have to do everything."* (Female caregiver from a focus group discussion, Za'atari).

Some brothers described how they still expected their sisters to fulfill their gendered labor roles but did so without imposing threats of physical violence. In Abala, for example, an adolescent girl shared that while she still must be on hand to support her brother when he returns from the market, she no longer fears and hides from him when he returns: *"Before we fought too much, he hit me, I hit him, now we have stopped, we have understood. He does not touch me and I also respect him. If I see him coming back from the market with luggage, I run to him to take the luggage and bring it home . . . Before, even if I see him with a lot of luggage, I only hid so that I did not have to come and help him."* (Adolescent girl from a paired interview, Abala).

Similarly, in Za'atari, an adolescent girl revealed that she is still expected to support her brother at home, but he has now refrained from physically punishing her if she does not want to comply: *"There were problems, for example, my brother used to hit me, and I used to curse him and such things Miss. But I respect him, and he feels for me now . . . For example, I once told my brother to fold his blankets and he said no. I swore that I wouldn't fold them, so he hit me. Now, if he tells me to fold them I decide if I will or I won't, and it will be ok."* (Adolescent girl from a paired interview, Za'atari).

3.1.3. Increased Emotional Connection and Social Support between Siblings

With the changes that adolescent girls in both study sites began to witness in their brothers' behaviors and attitudes towards them, girls also expressed that they felt more emotionally connected to their brothers. Some girls appreciated that their brothers began to *"tell [them] things about their lives . . . "*, and that they grew more emotionally responsive to them: *" . . . now if he sees me upset, he will calm me and ask me what's wrong . . . "* (Adolescent girl from a paired interview, Za'atari). Others began to view their sibling relationship through the lens of friendship. An adolescent girl living in a refugee camp in Abala described how her relationship with her brother is now akin to friendship; they no longer fight and argue but talk together and laugh:

> *"The program has affected my relationship with my brother because before we didn't talk like that, but now with our participation in the program as soon as he hears something he comes to me to talk, we talk, we talk, we laugh, we have become friends. And also before we only would fight, argue, we didn't even talk, everything that he tells me to do I would refuse to do because I don't respect it, but now we don't argue anymore, we talk, we chatter, we laugh."* (Adolescent girl from a paired interview, Abala).

Similarly, an adolescent boy in Za'atari described the trajectory of improvement in his and his sister's relationship, from isolated disengagement to engagement, connection, and kinship: *"For example, we have two parallel lines, the brother and the sister. They don't cross and each walks separately. Gradually, this program started to change their lines and make them lean towards one another till they met in a certain point, the point of agreement. They are one hand, they don't walk in separate lines anymore, they walk together."* (Male sibling from a paired interview, Za'atari).

Increased feelings of friendship and connection in turn impacted the sense of support that adolescent girls felt from their brothers. Given the innumerable risks faced by the participant girls from the very nature of their living within contexts of protracted conflict and displacement, being able to count their brothers among the number of trusted actors in their social networks seemed to have a positive impact on their sense of wellbeing. In Abala, for instance, adolescent girls spoke about leaning on their brothers for protection if they encountered community-level violence: *"Now I can tell my big brother if I see people fighting and I can't separate them, to go separate them."* (Adolescent girl from a paired interview, Abala) In Za'atari, an adolescent girl shared how she now felt she could go to their brothers for support if she encountered violence in their community, and moreover, she could trust that her brothers would believe her story:

> *"The brother has changed. He trusts his sister and shares his worries with her. If he faces a problem, he will tell her. Sometimes, a girl faced trouble in the street. A guy followed her, and someone saw that. The brother knew and he didn't trust his sister anymore. He suspected her and she didn't leave the house without him. The program teaches him that if something like this happened, he shouldn't suspect his sister. He should trust her and answer back to people who talk about her."* (Adolescent girl from a paired interview, Za'atari).

Other family members also observed their sons becoming trusted, supportive actors to their sisters. A female caregiver in Azraq shared that as her son progressed through SSAGE, he confronted some of the challenging realities that his sister would likely encounter in adolescence and adulthood and began thinking more critically about how his behaviors impact his sister's rights and freedom. He underwent a drastic shift– whereas he used to prevent his sister from going to school, feeling it was a burden to walk with her every morning, he began taking the responsibility of accompanying her to class very seriously. His mother shared:

> *"He has changed because he got information . . . that his sister shouldn't be pressured and deprived from education. He refused to take her to educational institutions because he didn't want her to go. Now, it is ok with him and when I tell him to take her because she*

*is going alone without her sisters, he would immediately change his clothes and willingly take her. He used to take her sometimes but unwillingly and only because I told him to.*" (Female caregiver from an in-depth interview, Azraq).

### 3.2. Pathways of Change in the Sibling Dyad

Our data show that changes to the sibling dyad were facilitated by two major factors: (1) male siblings' interrogation of masculinity alongside their peers in the SSAGE program and (2) improved communication between male and female siblings.

### 3.2.1. Male Siblings' Interrogation of Positive and Negative Masculinities

In Niger and Jordan, the SSAGE curricula for adolescent boys and male youth included sessions dedicated to exploring negative masculinities common in each setting and proposing more positive, reinterpreted masculine ideals. Male siblings in both settings seemed to be genuinely impacted by these sessions and were able to thoughtfully reflect on how negative masculinities and related gender norms showed up in their relationships with their sisters. Many expressed an interest in embodying more egalitarian masculine identities which prioritized open communication, emotional connection, and nonviolence. A great number of boys and male youth reflected that they no longer felt that " . . . *being a man is about hitting siblings to show them that I am a man.*" (Male sibling from a participatory research activity, Za'atari).

While adolescent boys and male youth across SSAGE sites shared various visions of positive, alternative masculinities, many centered on the premise that displays of dominance, aggression, and violence were undesirable attributes of masculine identities that they no longer wished to emulate. An adolescent boy from Azraq shared the following: "*Well, I heard that some people think that they act manly, show dominance and authority by hitting their sisters. I think it's the opposite. You should say kind words to your sister, buy her presents, ask her if she needs anything, and take her out to the garden for example.*" (Male sibling from a paired interview, Azraq). Similarly, a male sibling in Abala shared that he now thinks a true 'man' knows "*the value of women*" and does not use derogatory words and violent actions in his engagement with them. Instead, with increased awareness from the program, he chooses to show women and girls more respect: "*Regarding this SSAGE program, we know the value of women and this program has prevented us from hitting women, from saying bad words to them so we are all aware.*" (Male sibling from a paired interview, Abala).

These reinterpreted definitions of manhood discussed in the SSAGE sessions were incongruent with some of the boys' prior treatment of their sisters. Interestingly, some male siblings reflected on the stigma they anticipated they would experience if they performed household labor traditionally assigned to women or failed to assert positions of 'masculine' dominance over their sisters. The SSAGE sessions on masculinity addressed the stigma that adolescent boys may anticipate experiencing if they fail to live up to the dominant masculine ideals within their community; for some boys, this fostered greater self-awareness about the deleterious impact of negative masculinity on their relationships. This is exemplified in a self-reflection from an adolescent boy living in Za'atari, who found that he longer felt ashamed in seeking support from his sister following SSAGE participation: "*I used to feel embarrassed to ask for my sister's help. After participating in the program, and they gave me examples on that, I ask my sister about anything I don't know because it might affect her and me.*" (Male sibling from a participatory research activity, Za'atari).

Moreover, some participants In Abala, Za'atari, and Azraq also described the ways in which male siblings' evolved perceptions of masculinity, where men and boys no longer have to be figures of dominance and power, helped to facilitate improved sibling relationships. An adolescent girl in Abala described how brothers who participated in the intervention no longer use physical violence to display their strength and authority as young men but rather choose to advise and support them on issues that they face in

their daily lives: *"If we take before, our big brothers hit us even without any reason. But now with the arrival of "SSAGE" and because of the training, they made our brothers simply advise us on the different social mores."* (Adolescent girl from a paired interview, Abala). Caregivers also observed how changes in their son's expression of masculinity had positive impacts on their adolescent daughters. A mother in Azraq shared how changes in her son's understanding of masculinity and attitudes related to gender led to healthier communication, increased affective involvement, and heightened support and guidance between him and his sister:

*"I've seen that the relationship between my son and daughter have changed after the program. He shares things with her, tells her things, and helps her. He is supporting her now. They used to fight but now they have a good understanding. He assists her in studying and supports her inside and outside the house ... He thought that he was the guy and she was the girl and end of story. The program helped him change this idea and helped us become cooperative and participative. He helps her with studying and with matters inside and outside the house now."* (Female caregiver from an in-depth interview, Azraq).

### 3.2.2. Improved Communication between Siblings

Anecdotes from adolescent boys and male youth, as well as their female sisters and caregivers, suggest that improved communication between brothers and sisters facilitated male siblings' decreased violence perpetration and increased emotional connection and social support. One female caregiver in Azraq remarked on the changes in the quality of her children's communication with one another:

*"Mostly, the way my oldest son treats his sister [has changed]. He is more gentle and understanding now. If she tells him that she can't do something for him, he will tell her that he will do it himself. He is more understanding and compassionate now. I don't know [why these changes occurred] but it is related to the information they were given. Even the way she treats her brother has changed. They both have changed. She used to refuse to bring him things. Now she would tell him that she is tired and can't do things for him and she would ask him to do his stuff by himself. The way they talk has changed. They treat each other in a nicer way."* (Female caregiver from an in-depth interview, Azraq).

An adolescent girl's anecdote on changes she experienced in her life following the SSAGE program in Azraq reflects a similar story. She states the following about her brother: *"We started communicating more. We started talking to each other, telling each other everything that happens with us. If we felt sad, we went and told each other, as well as if we felt happy. If I face any problem I tell him. He [has] become a friend to me, not only my brother."* (Adolescent girl from a paired interview, Azraq). Adolescent girls in Abala also felt that improved communication between themselves and their brothers facilitated some of the greater changes in their relationships, such as a reduction in physical abuse. One girl shared the following: *"The program has changed our relationship because now my brother and I are together, we understand each other well whereas before we used to fight and insult each other but now everything has changed."* (Adolescent girl from a paired interview, Abala). Similarly, another adolescent girl in Abala found that she and her brother better share their views and work together to resolve disputes more peacefully following SSAGE participation: *"The program has affected our relationship with my brother well because now we are on the same page, we come and talk if we have a disagreement, and we find a solution."* (Adolescent girl from a paired interview, Abala).

Many adolescent boys also felt that improved communication paved the way for more respectful exchanges with their sisters, which fostered a greater understanding of their own needs, and a heightened ability to anticipate the needs of their sisters and other family members. One male sibling n Abala, for example, noted that with SSAGE participation, he better communicated and considered his sister's point-of-view: *"The most important thing that these trainings have strengthened between me and my sister, we understand each other, very well as I told you before we cannot collaborate with her and especially to discuss, but now that we*

*have followed the trainings, we understand each other perfectly.*" (Male sibling from a paired interview, Abala).

*3.3. Unanticipated Impacts of Male Siblings' Heightened Awareness to Protection Risks Faced by Their Sisters*

Despite the many positive impacts of the program on adolescent girls' relationships with their male siblings, there is some evidence that the SSAGE program content on girls' protection may have been interpreted by participants in ways that could negatively impact girls' rights and freedoms. In all program sites, a small minority of male siblings responded to their heightened awareness of GBV and girls' protection risks by imposing increased control and authority over their sisters, as a means of protecting them from violence. While brothers' increased interest in their sisters' safety and wellbeing can herald a positive change resulting in a deepening of girls' social support networks, it can also exacerbate existing gender norms, grant brothers further rights to control their sisters' behaviors, and reinforce girls' subordinate positions of power vis-a-vis their brothers. For example, this female caregiver from Za'atari reflected favorably on how her son has become more anxious about his sisters' movements throughout the community since his participation in SSAGE: "*My son . . . goes crazy when my daughter stands outside. He tells her to stay inside. He hangs out with guys his age and hears stuff that he doesn't like, and he worries about his sisters.*" (Female caregiver from a focus group discussion, Za'atari).

Restrictive interpretations of 'protection' were even more pronounced amongst SSAGE participants from the refugee and host communities in Abala. Several male siblings connected SSAGE learnings to their view that men should serve in the role of the 'benevolent protector' of women and girls. This male sibling from a host community in Abala, for instance, articulated that masculine ideals are premised on providing protection to women and girls: " *. . . This program is very important, because if someone wants to insult a girl, you have to protect her as a man, so soon she will know that you are a worthy and responsible man.*" (Male sibling from a paired interview, Abala). Another young man interpreted his role as 'protector' by correcting his sisters' actions that he thought were unsuitable: "*If you see your sister working hard you have to help her, if she's doing something wrong you have to correct her and tell her to avoid certain things that are not suitable.*" (Male sibling from a paired interview, Abala).

Other adolescent boys and male youth in Abala described their role as brothers to adolescent girls as "*heavy*" with responsibility, stating that they felt their role required them to keep their sisters "*on the right track*" as they transition from adolescence into adulthood. In the words of one male sibling living in a refugee camp, "*Now, if you have a sister at home that she's not married yet, that's a heavy load that you have. And then I, who am a man, I am better placed to put her on the right track.*" (Male sibling from a paired interview, Abala). A female caregiver corroborated this behavior by her son, and celebrated it:

> "*Now with the program we have known that with the accompaniment of a boy in a house can help to have an eye on the education of the girls, he is a protector because if he sees his sister in inappropriate places, he will make her come back home. Because a girl who hangs out everywhere can attract problems especially with the ill-intentioned men who will not hesitate to find a thousand ways to attract them with money or materials.*" (Female caregiver from an in-depth interview, Abala).

## 4. Discussion

This study explores the promise of SSAGE, a gender-transformative and sibling-inclusive family strengthening program, in mitigating gender-based sibling violence—an often-overlooked form of interpersonal household violence, which stems from patriarchal arrangements, disproportionately affects adolescent girls, and contributes to their onward risk of GBV and associated physical and mental health sequelae. Originally piloted in northeast Nigeria, and recently implemented among Syrian refugee families in Jordan and Malian refugees and nearby host community families in Niger, the SSAGE program is

novel in its explicit and simultaneous engagement of the whole family unit—inclusive of adolescent girls, their male siblings, and their male and female caregivers—to strengthen household relationships, reduce girls' immediate and long-term risk of violence, and enhance their protective assets in humanitarian settings.

Qualitative findings from Jordan and Niger suggest that addressing pathways of gendered sibling violence has a positive impact on factors associated with girls' wellbeing and protection. Short-term impacts include decreases in the perpetration of physical and verbal violence by male siblings, increases in perceptions of equity in the division of household labor between male and female siblings, and increases in emotional involvement and social support from male siblings to their sisters—all of which were self-reported or observed by participants and their immediate family members. These relational shifts between brothers and sisters are in line with well-evidenced risk/protective factors for adolescent girls living in humanitarian settings. These include: social and family support, which can impact girls' experience of IPV and other forms of VAWG; traditional gender roles, gendered social norms, and household power dynamics, which can impact girls' experience of IPV and child marriage; and family functioning, which can impact girls' mental wellbeing and resilience [44,62–66].

The Hoffman and Edwards integrated theoretical model is a useful framework that helps us understand more holistically the mechanisms by which the SSAGE program facilitated change. Drawing first from conflict theory, the model suggests that male and female siblings may use violence against each other to mitigate conflicts that stem from competing interests or resources (i.e., household labor and other familial responsibilities) and that sibling violence is mediated by verbal conflict. The qualitative data from our participants—adolescents and caregivers alike—reflect this reality and suggest that improved interpersonal communication helped facilitate some of the broader changes in sibling relationships, such as increased emotional connection, support, and household labor equity between siblings. This is in line with previous research on the SSAGE pilot in Nigeria [43,44] as well as other literature which finds that targeting verbal conflict and interpersonal communication can increase empathy and inspire emotional connection, which in turn helps to reduce conflict [24].

Turning to feminist theory, the Hoffman and Edwards model suggests that patriarchal arrangements at the structural, community, and familial levels, and ideals of masculinity work together to normalize VAWG and foster environments where various forms of household violence occur, including IPV and gendered sibling violence. Analysis of the SSAGE qualitative data suggests that many of the self-reported or observed changes in sibling dyad relationships were in part facilitated by adolescent boys and male youth interrogating the detrimental impact of negative masculinities on their relationships with their sisters, their broader families, their peers, and themselves. This finding is supported by various other research demonstrating that stress arising from men's perceived failure to conform to socially prescribed masculine gender roles and norms predicts their historical perpetration of IPV [67–69].

And finally, the Hoffman and Edwards model draws from social learning theory to describe the process by which siblings learn and model violent gendered interactions from their caregivers, peers, and the broader society they live in. Imitation, reinforcement, and an assessment of the probable rewards and punishments that one will receive for enacting certain behaviors are crucial steps by which gender-based sibling violence is learned within households. Our qualitative data suggest delivering the SSAGE program via synchronized age and gender-specific sessions for whole-family units enabled the diffusion of new ideas across peer and family groups; fostered the creation of new peer reference groups organized around positive masculinities and femininities; and created multiple terrains where individuals could try out new attitudes and behaviors and experience positive reinforcement.

Despite the positive shifts in sibling relationships, our data also suggest that in response to the program content on girls' protection, some male siblings reconceptualized

their role in their sisters' safety in ways that prioritized the continued control over girls' actions as a means of protecting them from violence. This is a challenging finding to interpret because the degree to which this approach protects girls from violence, or in fact furthers their risk of violence, depends on the lived realities and the threat of violence for girls in the public sphere in the given context. While a child protection or family functioning lens might interpret male siblings' increased attentiveness and oversight as a positive mechanism for girls' protection, a feminist analysis would find that brothers controlling their sisters' actions in the name of protection may protect girls from violence in the immediate term, but ultimately contributes to the ongoing diminishment of their agency, freedom, and rights in the long-term. Activists and scholars have called attention to the paternalism inherent in the protection framework that is commonly used in the humanitarian sector and its limitations for truly gender-transformative change [70–72].

The complicated findings from our research also highlight these limits and reveal the potency that behavior-change and norms-shifting programs can have in forming new, powerful reference groups in the lives of participating boys and men. Future whole family strengthening programs that aim to mitigate girls' risk of violence and improve their wellbeing must safeguard against protection messaging that can be taken up in unintended ways. Values clarification activities that differentiate protecting girls from violence and controlling or policing their behavior can also be integral to ensuring that program staff and participants understand the potential risk and are equipped to reframe the discussion. As a result of these findings, the SSAGE Implementation toolkit includes guidance on ways to facilitate protection messaging to reduce its risk of being interpolated in a restrictive and paternalistic way [73]. In addition, Mercy Corps has adapted the protection content in the SSAGE curricula so that the importance of listening to the experiences and needs of women and girls is highlighted as the foundation upon which men and boys should base their desire to provide support and protection.

Our research findings should be considered alongside a few key programmatic and study-related limitations. In the case of Jordan, quality control reviews of the study's qualitative transcripts against the audio files revealed some minor issues with transcription quality, which may have impacted the transcripts' accurate representation of the participants' full thoughts and opinions. In the case of Niger, research activities were truncated due to an increasingly volatile security situation in 2022, which made it unsafe for researchers and participants to gather to discuss the impacts of the program. In both sites, the self-reported or observed changes in participants following their SSAGE participation generally involved changes in attitudes, behaviors, and beliefs at the individual and household levels—not at the community level. Additionally, the changes reported in this paper derive solely from the qualitative component of the mixed-methods program evaluation. In both Jordan and Niger, the quantitative survey questionnaire was administered to relatively small samples of adolescent girls, male siblings, and adult caregivers and findings suggested limited improvements among program participants in Jordan and differences between participants and controls in Niger. Given that qualitative and quantitative data were collected approximately 1–3 months following completion of the SSAGE program, we also do not have insight into whether shifts were sustained in the longer term. Larger-scale mixed-methods evaluations of the long-term impacts of whole family strengthening programs are needed to better understand if changes at the individual and household level are sustained over time and extend into the broader community.

Another potential limitation to note is related to aspects of the SSAGE program design. While the program focuses explicitly on addressing household-level factors which can increase or mitigate girls' immediate and long-term risk of violence victimization, evidence on the association between household attitudes and adolescents' experience of violence is mixed, and often context-dependent. For example, a study exploring how individual and social norms are related to experiences of IPV among married adolescents in Niger finds that the extent to which individual attitudes versus social norms are associated with IPV experience differs between the adolescent female and adult male respondents [74].

Conversely, studies from other locations find that community-level norms are actually significantly associated with experiences of household violence [75]. This heterogeneity suggests that community-level norms and household-level attitudes have varying levels of influence over the experience of household violence, depending on the context. Future girls' protection programming using whole-family approaches should carefully consider the interplay of community norms and household attitudes to discern which levels should be prioritized in program designs.

## 5. Conclusions

The findings from this qualitative study offer contributions to various evidence bases. First, to the sibling violence literature which is heavily focused on high-income Global North settings, this study offers important insights into the experience of gendered sibling violence in humanitarian settings in the Global South. Second, this study contributes to existing evidence trends on adolescent girls' mental health and wellbeing in humanitarian contexts, further corroborating the importance of girls' social networks in mitigating their immediate and long-term risks of GBV and associated physical and mental health sequelae. Third, this study offers new insights for the research agenda on intersections between violence against women and violence against children; given the connection between a child's experience of caregiver-directed violence and their onward perpetration of violence against siblings (who are typically female and/or younger), sibling violence is a critical pathway of household violence that should not be overlooked. It is crucial that programmatic approaches which aim to both reduce household violence and mitigate girls' immediate and long-term risk of GBV target multiple relationships within the family, including that between male and female siblings as well as other context-relevant dyadic relationships where gender/power differentials play out. Longitudinal research on the impacts of targeting male-perpetrated sibling violence during adolescence on the future perpetration of violence against intimate female partners is needed.

**Author Contributions:** Conceptualization, A.K., M.G., J.D., I.S. (Ilana Seff) and L.S.; methodology, J.D., I.S. (Ilana Seff) and L.S.; validation, A.K., M.G., H.S., I.S. (Ibrahim Saley), A.M. and J.D.; formal analysis, A.K., M.G., K.A., H.S., I.S. (Ibrahim Saley) and A.M.; resources, H.S., I.S. (Ibrahim Saley) and A.M.; data curation, A.K., M.G. and J.D.; writing—original draft preparation, A.K. and M.G.; writing—review and editing, A.K., M.G., K.A., H.S., I.S. (Ibrahim Saley), A.M., I.S. (Ilana Seff), J.D. and L.S.; visualization, A.K.; supervision, A.K., M.G., I.S. (Ilana Seff), J.D. and L.S.; funding acquisition, J.D. All authors have read and agreed to the published version of the manuscript.

**Funding:** This research was made possible by funding support from the Government of Sweden through the Ministry for Foreign Affairs (Award No. UD2021/08942) and the United States Department of State through the Bureau of Population, Refugees, and Migration (Cooperative Agreement # SPRMCO20CA0162, CFDA # 19.522).

**Institutional Review Board Statement:** This study approved by the Jordan University of Science and Technology, King Abdullah University Hospital Institutional Review Board (Ref#24/142/2021) and the National Ethics Committee for Health Research of the Niger Ministry of Public Health (Ref# 40/2021/CNERS).

**Informed Consent Statement:** Informed consent was obtained from all subjects involved in the study.

**Data Availability Statement:** Not applicable.

**Acknowledgments:** We would like to thank the SSAGE study participants who shared their time and insights with us for this study and who participated in the SSAGE program. We would also like to thank Kevin McNulty (Mercy Corps), Dale Buscher (Women's Refugee Commission), REM Africa-Niger, and Consultus in Jordan for their contributions. Lastly, we would like to acknowledge the Women's Refugee Commission and Mercy Corps for their continued collaboration and support on the SSAGE implementations and adaptations.

**Conflicts of Interest:** The authors declare no conflict of interest. The funders had no role in the design of the study; in the collection, analyses, or interpretation of data; in the writing of the manuscript; or in the decision to publish the results.

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
