# Peer review of "The Brother–Sister Sibling Dyad as a Pathway to Gender-Based Violence Prevention: Engaging Male Siblings in Family-Strengthening Programs in Humanitarian Settings"

_adolescents, doi:10.3390/adolescents3010012_

Round 1
Reviewer 1 Report
The article deals with a topic that is highly relevant and worth publishing as it deals with the prevention of gender-based violence and the role of male siblings. The article also has an excellent structure. However, like any research paper, there are things that can be improved. Some ideas for improving your article are shared below:
About the theoretical framework.
· Although it is evident that you focused your topic on family dynamics, I think it is appropriate to include (in the first section) studies that explain how males who exert violence at home also suffer violence by their friends, in their social dynamics. The violent brother not only learns it from the father or mother, he also experiences violence and reinforces it with his friends and even with other male siblings who are older than him.
About the interview:
· Authors do not explain how they design the question guide for the in-depth interview.
· What questions did Authors include?
· How did Authors decide what was important to ask?
· Did Authors use the same interview question guide for all the different types of participants?
· Did Authors do any pilot testing to improve the guide?
· Can Authors include some questions from the guide?
About results:
· It is very common to include only success stories where the program worked to reduce gender-based violence against girls.
· The text omits cases where the program failed to improve the problem of violence or where improvements were partial. In my opinion, situations where programs do not work also provide valuable information for understanding the problem.
· In addition to the above, I recommend that as part of the analysis of results you can include a map showing the pathways of change.
· Using Qualitative Data Analysis (QDA) software could help authors to map qualitative information to demonstrate the pathway in which different concepts that were analyzed in the study are connected. For instance, how communication between siblings reduces violence.
About the conclusion:
· In my humble opinion, the conclusion should emphasize in a broader way how gender-based violence towards women in the family environment can be reduced through programs that support improving the relationship between siblings.
Reviewer 2 Report
1- Title is too long! should be changed!
2- This paper presents the results of the SSAGE and its effectiveness in solving or minimizing the violence against adolescents girls living in refugee camps. The results demonstrated that the successful based on interviewing people! in my opinion, we cannot based on just interviewing people to test the effectiveness of the program.
3- The study has been conducted on refugees in two different countries with different religion, different mentality, difference culture, different ethics... Authors did not study theses factors in the discussion part.
4- In addition, increase levels of unemployment, poverty, decrease levels of economic opportunity and community participation; poor housing conditions; gang activity, emotional distress and a lack of access to service! all of these issues exist in refugee camps, and affect and increase the violence against girls and women! Also, the decrease of violence, these issues and challenges should be solved. Authors did not highlight these factors in the paper.
5- The contents of this program is not clearly presented in the paper!
6- The paper missed some statistics about the kind of violence and the percentage of violence in the two studies places.
Reviewer 3 Report
The article is really well structured, clear and effective, properly referenced.
I think that an explanation of what patriarchy means in your understanding and how it is related to violence in general and GBV in particular is required.
A clarification of what you mean with participatory research activities (PARs) - line 236 wit a clear description.
I would also suggest a possible future step of the research considering how this program has affected male participants in their gender-based relationships with non-siblings women.
Round 2
Reviewer 2 Report
Authors updated the manuscript according to my comments. the revised manuscript is accepted